

# Biomechanical implications of skeletal muscle hypertrophy and atrophy: a musculoskeletal model

Andrew D. Vigotsky[1,2], Bret Contreras[3] and Chris Beardsley[4]

[1] Kinesiology Program, Arizona State University, Phoenix, AZ, United States of America
[2] Leon Root, M.D. Motion Analysis Laboratory, Hospital for Special Surgery, New York, NY, United States of America
[3] Sports Performance Research Institute New Zealand, Auckland University of Technology, Auckland, New Zealand
[4] Strength and Conditioning Research Limited, London, United Kingdom

Corresponding author
Andrew D. Vigotsky,
avigotsky@gmail.com

## ABSTRACT

Muscle hypertrophy and atrophy occur frequently as a result of mechanical loading or unloading, with implications for clinical, general, and athletic populations. The effects of muscle hypertrophy and atrophy on force production and joint moments have been previously described. However, there is a paucity of research showing how hypertrophy and atrophy may affect moment arm (MA) lengths. The purpose of this model was to describe the mathematical relationship between the anatomical cross-sectional area (ACSA) of a muscle and its MA length. In the model, the ACSAs of the biceps brachii and brachialis were altered to hypertrophy up to twice their original size and to atrophy to one-half of their original size. The change in MA length was found to be proportional to the arcsine of the square root of the change in ACSA. This change in MA length may be a small but important contributor to strength, especially in sports that require large joint moments at slow joint angular velocities, such as powerlifting. The paradoxical implications of the increase in MA are discussed, as physiological factors influencing muscle contraction velocity appear to favor a smaller MA length for high velocity movements but a larger muscle MA length for low velocity, high force movements.

## INTRODUCTION

Muscle hypertrophy is a common adaptation to mechanical loading, typically delivered in the form of long-term programs of resistance training. Muscle atrophy is a response to disuse that occurs quickly following even short periods of mechanical unloading, which can be as little as one week of strict bed rest (*Dirks et al., 2015*). Both muscle hypertrophy and atrophy have important implications for clinical, general, elderly and athletic populations, because of the relationship between measures of muscle mass or size and a range of performance and health outcomes. For example, among strength athletes, measures of muscle mass or size have been found to be very good predictors

of Olympic weightlifting, powerlifting, and strongman performance (*Brechue & Abe, 2002*; *Siahkouhian & Hedayatneja, 2010*; *Winwood, Keogh & Harris, 2012*). In the elderly, *Janssen et al. (2004)* found that low levels of muscle mass were strongly correlated with an increased risk of disability, *Malkov et al. (2015)* reported that reducing thigh muscle size was associated with an increased risk of hip fracture, and *Srikanthan & Karlamangla (2014)* found that low levels of muscle mass were associated with increased all-cause mortality. In various clinical populations, reports have been made of increasing mortality or re-hospitalization rates in individuals with lower levels of muscle mass (*Greening et al., 2014*; *Streja et al., 2011*; *Weijs et al., 2014*).

The relationships between muscle size, or more accurately muscle cross-sectional area (CSA), and measures of performance or disability are underpinned by the unique ability of muscle to produce force, with greater muscle CSA corresponding to greater force production. At the individual muscle level, force is produced by the contractions of single muscle fibers, which are collected together in groups, known as fascicles. Muscle CSA is measured in two ways: anatomical CSA (ACSA) and physiological CSA (PCSA). ACSA is the muscle CSA measured in the plane perpendicular to its tendons (the longitudinal axis), commonly recorded at the widest point along the muscle. PCSA is the muscle CSA measured perpendicular to the muscle fascicles, which can vary for different parts of a pennate muscle. The angle between the longitudinal axis and the direction of the muscle fascicles is the fascicle pennation angle ($\theta_p$). The component of force exerted by the muscle fascicles is $\cos \theta_p$, and therefore, as $\theta_p$ increases, this leads to a progressively smaller component of muscle force being transmitted along the longitudinal axis to the tendons. Consequently, where changes in muscle PCSA occur following a period of either mechanical loading or unloading, the effect on muscle force transmitted to the tendons will also be affected by any simultaneous changes in $\theta_p$, with the increases that occur following mechanical loading reducing the beneficial effect of hypertrophy, and the decreases that occur following unloading mitigating some of the adverse effects of atrophy (*Aagaard et al., 2001*; *Erskine, Fletcher & Folland, 2014*; *Kawakami et al., 1995*; *Seynnes, De Boer & Narici, 2006*). In order to accurately predict joint moments, it is important for musculoskeletal models to take into account all of the aforementioned architectural variables.

Such changes in muscle PCSA, ACSA, $\theta_p$, and fascicle length that follow from exposure to mechanical loading have been well documented (*Aagaard et al., 2001*; *Erskine, Fletcher & Folland, 2014*; *Kawakami et al., 1995*; *Seynnes, De Boer & Narici, 2006*). The changes that simultaneously occur in various measures of strength are often attributed to these architectural and morphological changes, in addition to neural adaptations (*Behm, 1995*). Yet, the change in muscle size appears to be one of the most important, if not the most important factor: *Erskine, Fletcher & Folland (2014)* reported that changes in both muscle volume and muscle ACSA were strongly associated with increases in maximum voluntary isometric contraction (MVIC) force following a 12-week period of elbow flexor resistance training. Although the MVIC force values reported by *Erskine, Fletcher & Folland (2014)* were recorded using a linear force transducer, they are actually indicative of a net joint moment. Nevertheless, it is important to note that measures of MVIC force recorded

externally using load cells (as in *Erskine, Fletcher & Folland (2014)*) are not identical to the isometric muscle forces produced internally, as the external force reflects a net joint moment. Internal isometric muscle force is a product of the single muscle fiber force (also called specific tension), PCSA and $\theta_p$. The internal isometric muscle force is transmitted along the longitudinal axis of the muscle through the tendons to act on the bones at either side of a joint, where it creates an isometric moment about the joint. This isometric joint moment is the product of the internal isometric muscle force and the moment arm (MA) length. The MA length is the perpendicular distance between the muscle line of action to the joint center of rotation. Consequently, changes in both internal isometric muscle force and MA length can affect the magnitude of the maximum isometric joint moment that can be produced.

Although muscle ACSA appears to be a key determinant of MVIC moment production, the changes in MA length that occur as a result of either hypertrophy or atrophy have been less well described. Nevertheless, it appears that there is a relationship between agonist muscle size and the MA associated with the joint action. *Sugisaki et al. (2010)* and *Akagi et al. (2012)* described a positive correlation between muscle size and muscle MA length. These findings suggest that larger muscles are likely to benefit from a longer MA and consequently that hypertrophy may lead to increases in MA length and atrophy to decreases in MA length, which may have impacts on the joint moment that go above and beyond alterations in muscle force. Indeed, *Sugisaki et al. (2015)* noted a small increase in triceps brachii MA length following hypertrophy of around 5.5%, albeit which they deemed to be small and possibly negligible.

Since the relationship between MA length and muscle size is not fully understood, the purpose of this paper is to develop a two-dimensional mathematical model to describe how changes in ACSA of the proximal elbow flexors change MA length, $\theta_p$, and joint moment contributions.

## MATERIALS AND METHODS

A mathematical model that related the ACSA of both BIC and BRA to elbow flexion joint moment contributions was created using WolframAlpha (Wolfram Research, Champaign, IL, USA) and Excel (Microsoft, Seattle, USA), and the regression was tested in Excel and Stata (StataCorp, College Station, TX, USA).

### Development of the musculotendinous unit

A position-elbow flexor ACSA hyperbolic cosine regression equation was extrapolated from *West et al. (2010)*, wherein magnetic resonance images (MRIs) were taken of the elbow flexors of 65 subjects, from 2.4 to 11.2 cm proximal to the elbow, in both elbow extension and in a neutral radioulnar joint position.

Let $x$ be distance from the elbow, from distal to proximal, in centimeters (cm). Let

$$\text{ACSA}(x) = -\cosh(0.48(x - 7.3)) + 23. \tag{1}$$

Therefore, the radius, assumed to be the centroid and average line of pull of the proximal elbow flexors, assumed to be a cylinder (*Van der Linden et al., 1998*), can be defined as a

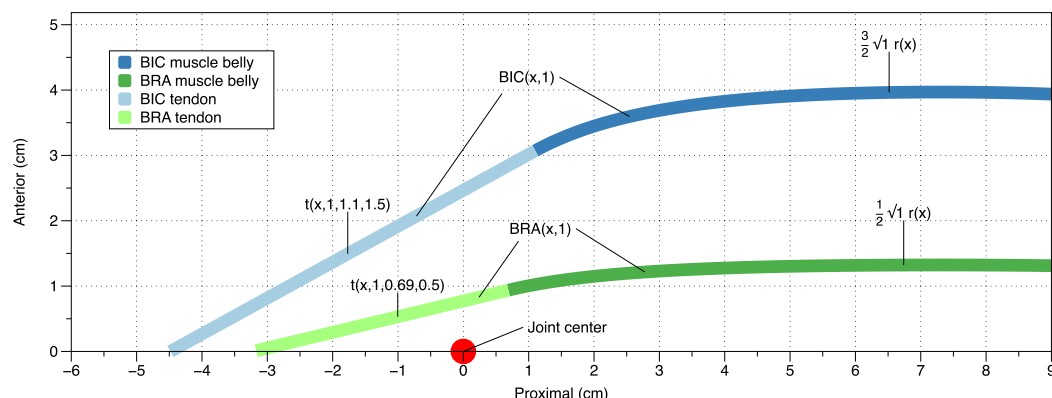

**Figure 1** **Graphical visualization of the model.** $x$ represents the proximal distance from the elbow joint. The muscle bellies end at the point where the tangent line, $t$, representative of the tendon, would insert into the appropriate location distal to the elbow. Moment arms were calculated as the perpendicular distance from the joint center (origin) to the tendon $t$. Parameters for $t$ represent the position proximal to the elbow, coefficient of hypertrophy, $x_{max}$, and anterior/superficial position. Parameters for BRA and BIC represent the position proximal to the elbow and coefficient of hypertrophy. The parameter of $r$ simply represents the position proximal to the elbow.

function of ACSA($x$):

$$r(x) = \sqrt{\frac{\text{ACSA}(x)}{\pi}}. \tag{2}$$

Let $\alpha =$ coefficient of hypertrophy, or the degree to which the muscle is hypertrophying (or atrophying) from baseline, which assumes uniform growth across the muscle belly.

Let $\beta =$ the $x$ position of the musculotendinous (MT) junction. The centroids of the distal BIC and BRA tendons, tangent to the centroid of the distal-most section of each muscle belly ($t$), assumed to be the line of pull, can be described as

$$t(x, \alpha, \beta, \varphi) = \frac{dr}{dx}(\beta)x + \varphi\sqrt{\alpha}r(\beta) - \beta\frac{dr}{dx}(\beta) \tag{3}$$

where $\varphi$ is a constant to correct for muscle location. More specifically, it was assumed that the BIC lay directly over (superficial and anterior to) the BRA. Because Eq. (1) represents the ACSA of both the BIC and BRA, the centroid of this (Eq. (2)) represents the division between the BIC and BRA, under the assumption that the muscles are equal in size. Therefore, when $\varphi = 0.5$, that represents half of the position of the centroid derived from Eq. (1), and when $\varphi = 1.5$, that represents one and a half times the position of the centroid derived from Eq. (1). Under the aforementioned assumptions, the former would represent the BRA, and the latter represents the BIC. The Cartesian outcomes of $t$ and $r$ in context of the model are illustrated in Fig. 1. The mathematical definitions of BIC and BRA depicted in Fig. 1 are described in the proceeding paragraphs.

Because $r(x)$ represents the centroid of both the BIC and BRA, it is necessary to divide this into each individual muscle. The BIC was set to begin 1.1 cm proximal to the joint center in order to control for insertion point, which was fixed 4.51 cm distal to the axis of

rotation (capitulum). This was assumed to be approximately the center of the insertion site, as the capitulum has a 1.06 cm radius (*Shiba et al., 1987*), the bicipital tuberosity is 2.5 cm distal from the radial head, and the insertion site is 2.2 cm long (*Mazzocca et al., 2006*). As per the MRIs, the muscle belly was set to end 11.2 cm proximal to the elbow joint. Therefore, the BIC MT unit (MTU) can be described as

$$\text{BIC}(x,\alpha) = \begin{cases} t\left(x,\alpha,1.1,\dfrac{3}{2}\right) & x < 1.1 \\ \dfrac{3}{2}\sqrt{\alpha}\,r(x) & 1.1 \le x \le 11.2. \end{cases} \tag{4}$$

The muscle belly of the BRA was set to begin 0.69 cm proximal to the joint center in order to control for insertion point, which was fixed 3.17 cm distal to the axis of rotation (trochlea). Like the BIC, it was assumed that this was the center of the insertion site, as the trochlea has a 0.75 cm radius (*Murray, Buchanan & Delp, 2002*), the coronoid process is about 1.10 cm from the trochlea, and the insertion site is about 2.63 cm long (*Cage et al., 1995*). Therefore, the MTU of the BRA can be described as

$$\text{BRA}(x,\alpha) = \begin{cases} t\left(x,\alpha,0.69,\dfrac{1}{2}\right) & x < 0.69 \\ \dfrac{1}{2}\sqrt{\alpha}\,r(x) & 0.69 \le x \le 11.2. \end{cases} \tag{5}$$

Both the BIC and BRA were modeled on the interval $0.5 \le \alpha \le 2.0$, with a step of 0.1.

## Moment arm, force, pennation angle, and normalized muscle force calculations

The joint center of the elbow is represented by the origin $(0, 0)$, and the perpendicular distance from the tendon to the joint center was then calculated as the MA. This was done so by finding the angle of insertion *via* arctangent and using that angle to find the vertical, or perpendicular, component by multiplying the lever arm by the sine of the insertion angle (Eq. (6)).

$$\text{MA} = \gamma \cdot \sin\left(\arctan\left(\frac{\partial\,(\text{BIC} \vee \text{BRA})}{\partial x}(\beta,\alpha)\right)\right) \tag{6}$$

where $\gamma$ is the muscle's insertion point.

Increases in pennation angle were assumed to occur proportionally with increases in maximum ACSA (*Erskine, Fletcher & Folland, 2014*). In order to calculate muscle force, normalized muscle force (NMF) was assumed to be 30.75 N cm$^{-2}$, which is the average of previously reported values (23, 30, 33, and 37 N cm$^{-2}$) (*Edgerton, Apor & Roy, 1990*; *Ikai & Fukunaga, 1968*; *Nygaard et al., 1983*; *Ralston et al., 1949*), and was assumed to remain constant with changes in ACSA. Muscle force ($F_{\text{muscle}}$) was derived from NMF by multiplying NMF by the ACSA (Eq. (7)). The use of ACSA rather than PCSA is appropriate in this context, as *Kawakami et al. (1994)* found no statistical differences between ACSA

**Table 1 Anatomical cross-sectional areas, moment arm lengths, and moment contributions.** Taking into account the changing moment arm broadens the elbow flexion moment contribution range of both the biceps brachii and brachialis.

|  | Atrophy | $\Delta_{AB}$ | Baseline | $\Delta_{HB}$ | Hypertrophy |
|---|---|---|---|---|---|
| BIC ACSA (cm$^2$) | 5.50 | −5.50 | 11.00 | +11.00 | 22.00 |
| BRA ACSA (cm$^2$) | 5.50 | −5.50 | 11.00 | +11.00 | 22.00 |
| BIC MA length (cm) | 1.63 | −0.54 | 2.17 | +0.59 | 2.76 |
| BRA MA length (cm) | 0.54 | −0.21 | 0.75 | +0.28 | 1.03 |
| BIC moment contribution (N m) (incl. MA) | 2.73 | −4.34 | 7.07 | +9.10 | 16.17 |
| BRA moment contribution (N m) (incl. MA) | 0.9 | −1.59 | 2.49 | +4.05 | 6.54 |
| BIC moment contribution (N m) (excl. MA) | 3.63 | −3.44 | 7.07 | +5.62 | 12.69 |
| BRA moment contribution (N m) (excl. MA) | 1.26 | −1.23 | 2.49 | +2.27 | 4.76 |

and PCSA of the elbow flexors.

$$F_{\text{muscle}} = \text{ACSA} \cdot \text{NMF}. \tag{7}$$

Tendon force ($F_{\text{tendon}}$) was then calculated as the parallel component of the muscle's force (Eq. (8)).

$$F_{\text{tendon}} = F_{\text{muscle}} \cdot \cos\theta_p. \tag{8}$$

### Moment contributions

The elbow flexion moment contributions of the BIC and BRA ($M$) were calculated by simply multiplying each muscle's tendon force by its respective moment arm.

$$M = F_{\text{tendon}} \cdot \text{MA}. \tag{9}$$

### Validation

In order to validate the model, the primary outcomes—that is, MA length, pennation angle, and joint moment trends—were compared with previous, relevant literature.

## RESULTS

Equation (1) showed a strong correlation with the length-ACSA relationship described by *West et al. (2010)* ($p < 0.001$; $r = 0.911$). The results of the model, including the effects of ACSA on MA, elbow flexion moment contributions, pennation angle, muscle force, and tendon force, can be found in Fig. 2. The key data for the atrophy, baseline, and hypertrophy conditions are shown in Table 1.

BIC and BRA MA lengths increased approximately hyperbolically with increases in ACSA (Fig. 2A). Angle of pennation for both BIC and BRA increased linearly with increases in ACSA (Fig. 2B). Both the BIC and BRA were shown to have identical muscle forces (Fig. 2C), but due to the differences in pennation angles (Fig. 2B), the differences in tendon force became more apparent with greater muscle ACSA (Fig. 2D). Figure 2E depicts how changes in MA length affect elbow flexion moment contributions, as the dashed

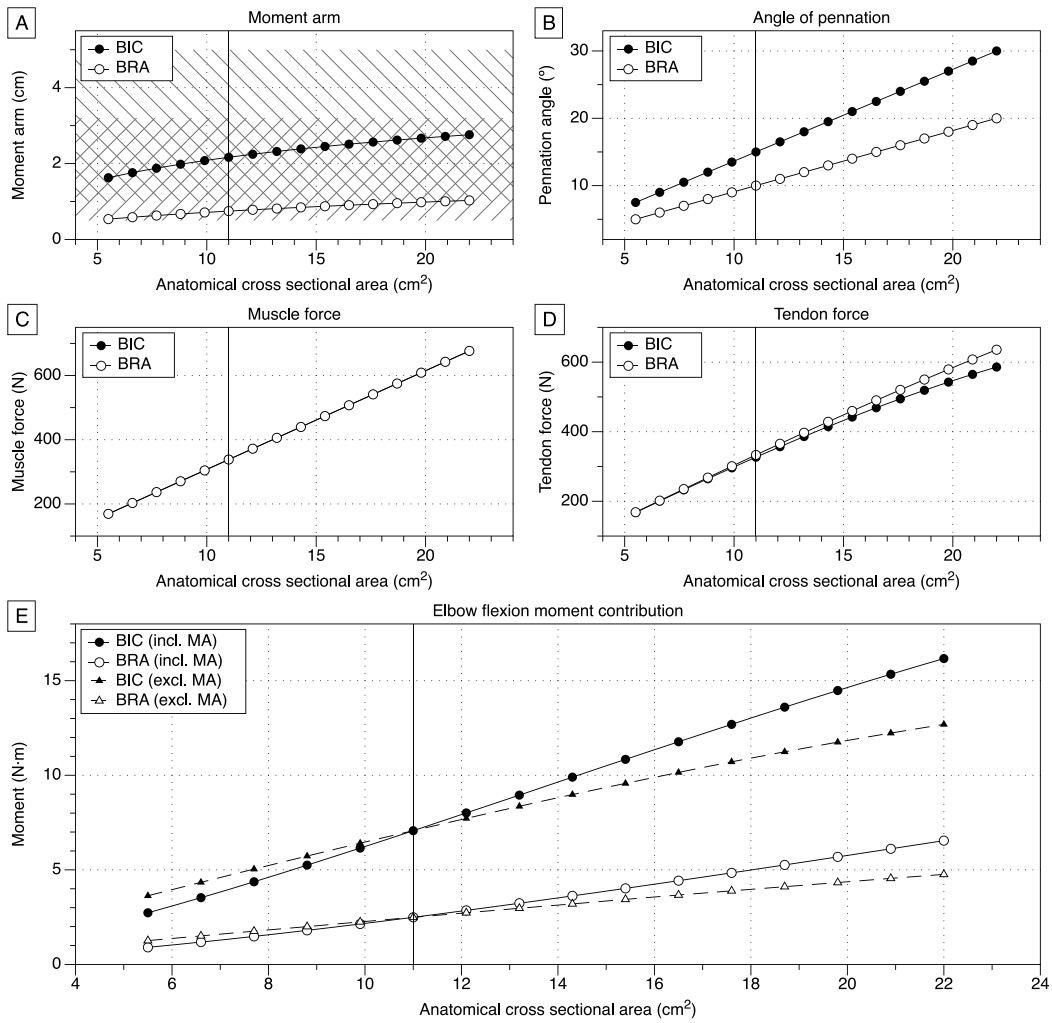

**Figure 2 Biomechanical variables as a function of anatomical cross-sectional area.** (A) Relationship between biceps brachii anatomical cross-sectional area and muscle moment arm. Negatively sloped lines are normal BIC MAs, and positively sloped lines are empirical BRA MAs (*Ramsay, Hunter & Gonzalez, 2009*). (B) Pennation angles of the BIC and BRA increase linearly with increases in ACSA. (C) Muscle force increases with ACSA. (BIC plots are underneath, and identical to, the BRA plots.) (D) Tendon force increases with ACSA. (E) The elbow flexion moment contributions of the BIC and BRA with changes in ACSA, both with and without the changes in MA length. Vertical lines at 11.0 cm² indicate baseline.

lines illustrate the moment contributions if the MA length did not change (remained identical to baseline (ACSA = 11.0 cm²)). At 22.0 cm², accounting for changes in BIC and BRA MA length result in 27.2% and 37.3% greater joint moment contributions, respectively (Fig. 2E).

## DISCUSSION

The MA lengths of the BIC and BRA at baseline in this model were within previously reported ranges in some studies (*An et al., 1981*; *Ramsay, Hunter & Gonzalez, 2009*), and although they may appear shorter than those reported by some other investigators (*Amis, Dowson & Wright, 1979*; *An et al., 1981*; *Edgerton, Apor & Roy, 1990*; *Pauwels, 1980*), most

studies did not report MA lengths in the same joint positions utilized for this model. Furthermore, the MA lengths reported in this model are similar to those previously modeled by *Murray, Delp & Buchanan (1995)* (Fig. 2A). The increases in pennation angles reflect values and trends previously reported, in that pennation angle increases linearly with increases in ACSA (*Erskine, Fletcher & Folland, 2014*; *Ikegawa et al., 2007*; *Kawakami, Abe & Fukunaga, 1993*; *Kawakami et al., 1995*) (Fig. 2B). Training studies corroborate the described increase in elbow flexion moment contributions, as it has been shown that net elbow flexion moments increase linearly with increases in ACSA (*Erskine et al., 2010a*; *Erskine et al., 2010b*).

To the authors' knowledge, this is the first model to describe the effects of muscle hypertrophy and atrophy on MA length. It revealed important changes in MA lengths of the BIC and BRA with increases in ACSA (Figs. 2A and 2E). Previous research has only attributed increases in net joint moment production to the effects of hypertrophy on muscle force (*Aagaard et al., 2001*; *Erskine, Fletcher & Folland, 2014*), while ignoring potential changes in MA length, as described by our model. Intuitively, this change in MA length must be a function of a change in insertion angle, as the insertion point cannot shift. This increase in insertion angle occurs as the size of the muscle belly increases, thereby shifting the resultant vector of the muscle farther from the humerus and consequently from the joint center. This is exemplified by the ACSA at baseline ($11.0 \text{ cm}^2$) corresponding to MA lengths of 0.75 cm and 2.17 cm for the BRA and BIC, respectively, while the ACSA of a muscle double that size ($22.0 \text{ cm}^2$) corresponds to MA lengths of 1.03 cm and 2.76 cm for the BRA and BIC, respectively (Table 1, Fig. 3). In other words, for a 100% increase in ACSA, the MA lengths of the BIC and BRA increase by 27.2% and 37.3%, respectively.

The modeled change in MA length for the BIC and BRA in this particular joint position (i.e., neutral radioulnar joint in elbow extension) is proportional to the arcsine of the square root of the change in ACSA ($\Delta \text{MA} \propto \arcsin(\sqrt{\Delta \text{ACSA}})$). This is due to lengths being proportional to the square root of the area in which they are contained (as in Eq. (2)), while the arcsine controls for the perpendicular component (as in Eq. (6)). The relationship observed by *Sugisaki et al. (2015)* elucidates just how subtle changes in MA length can be, as a 33.6% increase in triceps brachii ACSA was accompanied by a 5.5% increase in MA length. It is unclear if other muscles, joints, or joint angles would behave similarly under hypertrophy, as these results cannot be extrapolated.

Although the increase in MA length appears to be beneficial for static or quasi-static strength, it may be detrimental to high velocity, dynamic movements. This is paradoxical, because a larger MA length should be helpful in producing a larger net joint moment, which might be expected to increase angular acceleration, irrespective of joint angular velocity. However, owing to the biomechanical properties of muscle, this is not the case. Specifically, in accordance with the hyperbolic force–velocity relationship, less force can be developed during a high velocity, dynamic muscle action than during a slow velocity or isometric muscle action (*Hill, 1938*). It is thought that the force–velocity relationship may arise as a result of a number of factors, including the number of cross-bridges (*Piazzesi et al., 2007*), fluid friction or viscosity (*Gasser & Hill, 1924*; *Hill, 1922*), adenosine

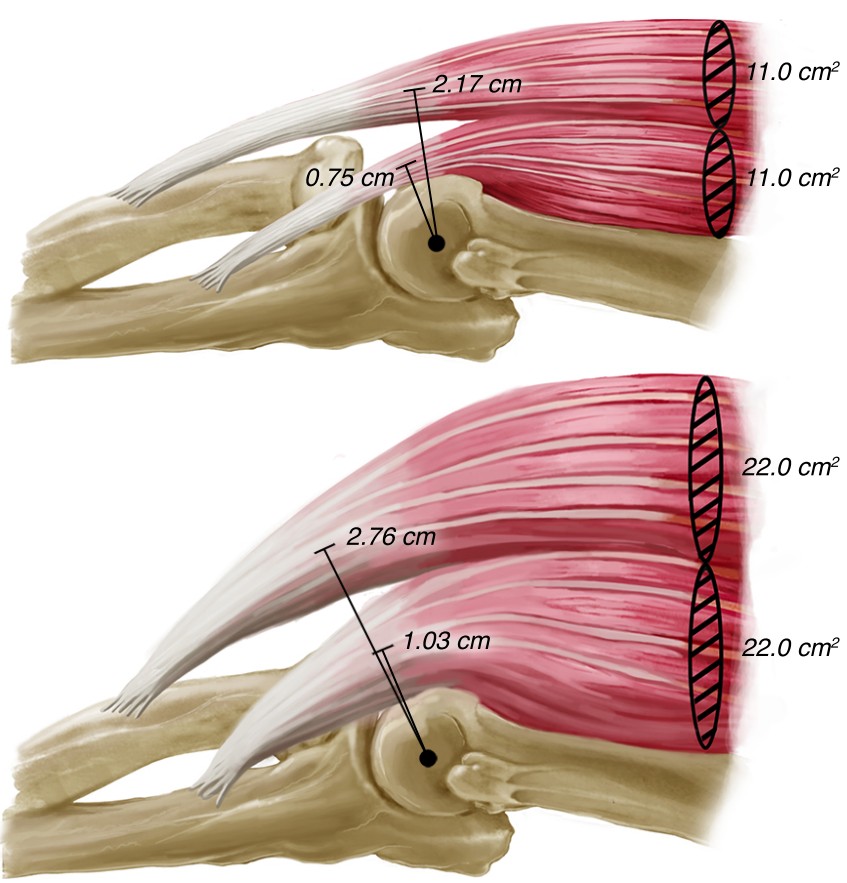

**Figure 3 Illustration of the changes in biceps brachii and brachialis moment arm lengths with increases in anatomical cross-sectional area.** By doubling the anatomical cross-sectional area of the biceps brachii and brachialis, the moment arms of each increase by 27.2% and 37.3%, respectively.

diphosphate (ADP) dissociation rates (*Nyitrai et al., 2006*), and passive elastic tension at long lengths (*Abbott & Wilkie, 1953*). Irrespective of the underlying mechanisms, however, *Nagano & Komura (2003)* showed that a shorter MA length is beneficial for high velocity, dynamic muscle actions, since a given length change in the muscle will necessarily cause greater joint excursion. Consequently, with a shorter MA length, more muscle force can be produced, as a shorter change in length is required, which necessitates a smaller contraction velocity. Mathematically, this can be described by the equation for MA length, where *dl* is the instantaneous change in MTU length and *dθ* is the instantaneous change in joint angular displacement (in radians) (Eq. (8)). With a larger MA length, a greater change in MTU length must occur for the same change in joint angular displacement, which would require a greater contraction velocity. The interactions of the ramifications of an increased MA on angular velocity and angular acceleration require further investigation, and may have several, important sport- or task-specific implications.

$$\mathrm{MA} = \frac{dl}{d\theta}. \tag{10}$$
## LIMITATIONS

There were a number of limitations inherent in this study that arose from the assumptions that were made. It was assumed that there was no change in either muscle fascicle or tendon length during the production of the calculated contributory elbow flexion joint moments, but changes in both muscle fascicle length and tendon elongation have been observed *in vivo*, in the tibialis anterior (*Ito et al., 1998*). Similarly, it was assumed that no myofascial force transfer occurred between the elbow flexors, although it has been observed that there are several ways in which this could occur (*Huijing & Jaspers, 2005*).

Our model assumed no changes in NMF, which directly affects muscle force calculations and thus joint moment contributions. Although force production typically increases to a greater extent than ACSA, some studies have reported reductions in normalized force following resistance training, albeit not in the elbow flexors (*Ikegawa et al., 2007*; *Kawakami et al., 1995*; *Sale, Martin & Moroz, 1992*). When considering studies performed in the elbow flexors, it is, however, apparent that normalized force typically either increases (*Brandenburg & Docherty, 2002*; *Vikne et al., 2006*) or remains constant (*Narici & Kayser, 1995*; *Takarada et al., 2000*). These equivocal findings indicate that our model may or may not reflect the typical changes in normalized force expected during resistance training for the elbow flexors, which may therefore arise due to central factors influencing strength gains or peripheral changes affecting single fiber specific tension.

Indeed, although excluded in our assumptions, studies have shown increases in single fiber specific tension using both *in vivo* (*Erskine et al., 2011*; *Erskine et al., 2010a*; *Erskine et al., 2010b*) and *in vitro* designs (*Erskine et al., 2011*; *Pansarasa et al., 2009*; *Parente et al., 2008*) following resistance training protocols, which were not necessarily accompanied by muscle hypertrophy. Changes in specific tension observed *in vivo* indicate that changes in either extracellular lateral force transmission or myofibrillar packing density might be responsible for alterations in normalized force, while changes observed *in vitro* can likely only be explained by alterations in myofibrillar packing density. Additionally, it has been reported that changes in specific tension occur in tandem with atrophy and that such changes appear to be associated with reduced myofibrillar packing density (*Riley et al., 1998*). Whether the exact same mechanisms are involved in the alteration of specific tension with mechanical loading and unloading, however, is unclear.

The use of a cylinder to model the BIC and BRA is not necessarily completely accurate, but cylindrical shapes have been used in previous muscle models, such as the gastrocnemius (*Van der Linden et al., 1998*). However, notwithstanding this point, the same proposed mathematical relationship would still stand with other, irregular shapes, as all linear distances within a cross-section are related to the square root of the area of that cross-section.

Our model assumed that there were no changes in fascicle length, changes in fiber type area proportion, or shifts in the fiber type of individual muscle fibers. However, previous research indicates that changes do occur in fascicle length (*Baroni et al., 2013*; *Blazevich et al., 2003*; *Reeves et al., 2009*) and in respect of proportional muscle fiber type areas (*Campos et al., 2002*; *Schuenke et al., 2012*; *Staron et al., 1991*), although the literature is currently

equivocal regarding the exact nature of these responses to both resistance training and disuse atrophy (*De Souza et al., 2014*; *Kawakami et al., 1995*). In addition, the effect of muscle fiber type on maximal strength is likely less important than its effect on dynamic strength, as the specific tension of type I and type II muscle fibers is not normally found to be substantially different (e.g., *Harber & Trappe, 2008*) but there is often a marked difference reported in muscle contraction velocity (e.g., *Harber & Trappe, 2008*). However, it can be argued that the model is more robust and free from possible confounding factors because the aforementioned variables were not included, so they do not have to be "teased out."

Additionally, although excluded in our assumptions, changes in agonist voluntary activation or co-contraction might be expected to occur following periods of mechanical loading or unloading. Indeed, some studies have reported small increases in voluntary activation following resistance training (*Ekblom, 2010*; *Erskine et al., 2010b*), which may imply that mechanical loading involves changes in neural drive. Whether such increases are likely to have a substantial impact on strength gains given that voluntary activation levels of >93% are frequently observed in young, untrained subjects before commencing resistance training (*Erskine et al., 2010b*; *Power et al., 2015*; *Venturelli et al., 2015*) appears questionable. Nevertheless, there is some evidence that voluntary activation is reduced in the elderly, possibly following sustained periods of disuse leading to atrophy (*Klass, Baudry & Duchateau, 2007*), and there is good evidence that such reductions in voluntary activation can be reversed following sustained programs of resistance training (*Arnold & Bautmans, 2014*). The effect of changes in antagonist co-contraction following periods of mechanical loading and unloading is much less clear. There does not appear to be any difference between antagonist co-contraction activity between trained and untrained individuals (*Maeo et al., 2013*), nor does antagonist co-contraction activity change following training in young (*Maeo et al., 2014*) or old (*Arnold & Bautmans, 2014*) individuals. Like the changes in fascicle and fiber characters, it can also be argued that by not including these neural adaptations, the model includes less possible confounding variables. Therefore, the modeled changes in forces and moments are strictly due to the architectural characteristics included in the model.

Lastly, and perhaps most importantly, this model only examined two muscles in one joint position (i.e., neutral). It is likely that with changing joint positions, the relationship would shift (*Murray, Delp & Buchanan, 1995*). For example, during elbow flexion, BIC and BRA MA lengths are greater because the insertion angle approaches 90°, so greater hypertrophy may be disadvantageous for MA length in such positions, as it may shift the insert angle away from 90°. However, such circumstances have yet to be described and modeled.

## CONCLUSIONS

The contribution of changes in ACSA to joint moment contributions following hypertrophy and atrophy resulting from mechanical loading and unloading are not fully understood, nor are the implications of concomitant changes in MA length. This model was the first to describe how changes in ACSA following hypertrophy or atrophy of the BIC and BRA might alter MA length and how both changes in ACSA and MA length

# PeerJ

might impact on relative elbow flexion moment contributions in neutral radioulnar and elbow joint positions. The results of this model should be interpreted with caution, as the predicted outcomes (namely, MA lengths) have not been demonstrated *in vivo*. Additionally, only one joint position (neutral) was investigated on two muscles, so the results cannot be extrapolated to other muscles or joint positions. Nevertheless, this model may serve as an effective tool for generating hypotheses that may inform experimental research with biomechanical implications.

## ACKNOWLEDGEMENTS

We would like to thank Professor Stuart Phillips, for providing the position-CSA data necessary to complete this model, and Dr. Silvia Blemker, for reviewing and critiquing an early rendition of our model.

### Funding

The authors received no funding for this work.

### Competing Interests

The authors declare there are no competing interests. Chris Beardsley is an employee of Strength and Conditioning Research Limited.

### Author Contributions

- Andrew D. Vigotsky conceived and designed the experiments, performed the experiments, analyzed the data, contributed reagents/materials/analysis tools, wrote the paper, prepared figures and/or tables, reviewed drafts of the paper.
- Bret Contreras conceived and designed the experiments, wrote the paper, reviewed drafts of the paper.
- Chris Beardsley wrote the paper, prepared figures and/or tables.

### Data Availability

We have included the dataset in Supplemental Information.

### Supplemental Information

Supplemental information for this article can be found online at http://dx.doi.org/10.7717/peerj.1462#supplemental-information.

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
