# Peer review of "Biomechanical implications of skeletal muscle hypertrophy and atrophy: a musculoskeletal model"

_PeerJ, doi:10.7717/peerj.1462_

## Round 0.1 · original submission · Major Revisions

We have fairly swiftly obtained two detaied, constructively critical reviews of the paper. It is very clear that major revisions are needed and re-review. However, the reviewers agree that this study has the potential to be a good publication once suitably revised. Ensure to address all points individually in a Response document, please.

·

Basic reporting

The authors state as their primary goal to investigate how muscle hypertrophy/atrophy may change muscle moment arms. In the introduction, this question is placed within the context of “strength”. However, as the authors acknowledge, there are many different measure of strength. For example, strength could be defined as the maximum isometric strength of a single muscle or the maximum isometric joint moment. How these different measures of strength relate to each other and the question of interest is not clearly presented in the introduction. For example, a key concept that the reader needs to understand in order to comprehend the underlying purpose of the study is how traditional measures of joint strength (e.g., isometric joint torques) are related to muscle force and moment arms. The authors also discuss different other muscle specific measures such as ACSA and PCSA. However, how all of these link together would not be clear to the general reader. A revised introduction that clearly introduces and defines these concepts and their relationships would greatly improve a reader’s understanding of the importance of the research question being addressed. (e.g., ACSA/PCSA -> maximum isometric force, Fiso*MA = Max Muscle Moment, etc.).
The authors also seem to use ACSA, PCSA and a general acronym CSA interchangeably throughout the manuscript. PCSA and ACSA have very specific meaning in the literature and it is important to distinguish between the two quantities. CSA appears to be used mainly as ACSA. However, the use of CSA is not necessary in the manuscript. I suggest the authors go through and remove CSA, replacing it with ACSA/PCSA as appropriate. For example, Eq 1 uses CSA, but then first paragraph in the results refers to Eq. 1 as being and ACSA-length relationship).
The aim of the manuscript is also vague as currently stated in the introduction. I recommend a focused aim on the most relevant question (ACSA vs. MA), which help provide a clear focal point for the introduction and discussion. For example, the aim could be rewritten as “…the purpose of this paper is to develop a two-dimensional mathematical model to describe how changes in ACSA of the proximal elbow flexors change the muscle moment arm lengths”. This is clear and concise. The rest of the information in the sentence, as currently written, relate to implications to the results. The study does not directly assess elbow flexion joint moments – the model does not contain all of the necessary muscle to do this nor does it need to.
The authors also spend a large amount of time discussing model limitations. These limitations first appear in the materials and methods and then three pages of discussion are devoted to model limits. Good modelling practice is to generate a model only complex enough to address the question of interest. I feel the authors have done so (ACSA versus MA). There is no need for their model to include other things such as force-length-tension relationships, activation patters, etc., as they likely will not influence their findings. However, these model limitations do make for interesting discussion points when extrapolating or speculating on the key study finding of ACSA-MA relationships to joint level measures and should be discussed in this context.
I also believe that the current set of figures is insufficient for the study. An additional figure showing a representation of the author’s 2D model, with the key variables of interest would be very useful when going through the model equations. Many of the equation variables could also be clearly defined using a figure. For example the authors use x as the distance from the elbow joint centre. However, in what direction is that distance? Along the humerus? Along the muscle belly or tendon line of action? A figure would greatly clarify these variables without the reader being forced to make assumptions.
Figure 1 should be broken up into multiple figures and discussed independently and in more detail. The results section states “The results of the model, including the effects of ACSA on MA, normalized muscle force… can be found in Figure 1”. This is the extent of the description of the results presented in the figures. The authors should “walk the reader” through each figure, explaining clearly dependent and independent variables and general/notable trends.
Figure 2 is very useful. However, the moment arm lengths labelled do not appear to be presented or identified in the text of the manuscript. If values are to be presented in a figure then they should be highlighted in the text. Also, this figure may be a good place (or the model figure) showing some of the modelling parameters.

Experimental design

As this is a modelling study, there is no empirical design required. With the exception of some details, the authors have provided a reasonable overview of their model. I would strongly recommend, on publication, of including the model in supplemental information or in some other repository.
The key weakness to the model is the lack of model validation. The authors have acknowledged this weakness and I am familiar with the difficulty of validating muscle level models using grosser experimental data. However, the authors have done some high-level validation (e.g, comparing muscle joint moment arms to previous models, extrapolating joint level moments to compare against previous experimental studies). These attempts at validation should be presented clearly. The methods should have a section that clearly states how they are analysing their model (i.e., how are you generating the data for your figures/plots) and how they are attempting to validate the model. For example, in the methods the authors could state, “To validate our model, MA results were compared with those in the literature..”. I then recommend reorganizing the discussion to include a section focused on the model validation results.
Specific comments:
Equation 1: what are the units for the variables? cm, mm? Be clear in this equation and those following
Equation 2: What is a “force vector field”? In this equation the authors have assumed that muscles are circular and use the equation of the area of a circle to relate radius to ACSA. This assumption should be clearly stated as it influences the study findings.
Equation 3: What exactly is t? The inputs are clearly defined but what t actually represents is not clear. As mentioned above, a figure showing this and other variables would be very useful.
Equation 4 and lines 116-117. The authors appear to be switching between units of mm and cm. Please be consistent.
Equation 4, Equation 5: In both these equations a term is introduced on the muscle side 1.5sqrt(alpha) or 0.5sqrt(alpha). Where does this term come from?
Equation 6: understanding how this equation was generated is not clear to me. Some additional explanation would be useful.
Equations 7-10: The authors have chosen to determine muscle force from a predetermined joint moment using the relative contributions of the two muscles (as estimated from previous studies). Why did the authors choose this approach? It makes more sense to me to use an alternative approach where one would just assume that the muscles generate a maximum isometric force and then calculate the resulting muscle contributions to the joint moment. To me this would be more straight-forward and eliminate some of the assumptions that are required to transition from joint level to muscle level data. With your model, you would still be able to distinguish between muscle joint moment increase due to increase in PCSA alone versus contributions from changes in MA.
Equation 9: This equation is incorrect as stated and may explain much of what the author’s state as model inaccuracies in the discussion. As stated, the equation suggests that, at pennation angles of 90 degrees, the muscle force required to produce any tendon force is 0. The correct equation is Ftendon = Fmuscle * cos(theta). See: Zajac, FE (1989). Muscle and Tendon: Properties, Models, Scaling and Application to Biomechanics and Motor Control. Critical Reviews in Biomedical Engineering. 17(4). Pgs 359-411 for details on this derivation.

Validity of the findings

I do believe that changes in MA as a result of changes in muscle volume could have important consequences. The authors cite previous work suggesting that 5% increases in MA may occur. I agree with the authors, that in some movements, a 5% change in muscle-joint gearing could be biologically relevant.
The authors also clearly indicate that the model may not generalize to other muscles. Indeed, I think this is an important point as muscle anatomy will play a very large role as to how muscle might change at each joint.
The authors also suggest that the moment arms are proportional to the square root of ACSA. However, I believe that proposing this relationship is quite speculative as the relationships appears to come directly from the authors assumption that the ACSA is a circle, as the linear measure (radius) is related to the area measure (ACSA). As such, the result appears to be more due to the model definition and not as some sort of emergent, independent result. I imagine that this relationship could change dramatically if underlying assumptions in the model or the muscle geometry were changed, which would be an interesting way to test some of the model assumptions.
There also appears to be an error in one of the equations that was used to generate some of the figures, producing erroneous results.

Additional comments

This manuscript presents an interesting study that, in essence, is sensitivity study. The goal of the study was to determine the how changes in muscle mass/anatomical cross sectional area (ACSA) influence individual muscle moment arms and, therefore, maximum joint moments. The question addressed would be of interest to a broad range of researchers, including groups studying movement rehabilitation, sport performance and musculoskeletal system dynamics. To address the question of interest the authors have developed a musculoskeletal model. I believe that the model developed incorporates all of the features required to answer the study question. However, I do have some general concerns related to the manuscript in its current form, which I have provided in the other review sections.

Reviewer 2 ·

Basic reporting

No comments.

Experimental design

Eq. 2 assumes that the muscle is cylindrical. Also assumed that BIC lay directly over BRA (lines 104-105). This assumption wasn't noted in the discussion section but it seems like it would substantially impact moment arm calculations. How reasonable is the assumption of cylindrical muscle?

Line 100, "collinear" or other similar term may better describe the geometry than "tangent".

Line 100, use "superficial" rather than "distal", since "distal" is generally used anatomically for location further along the upper limb toward the forearm and hand.

Equation 3, please clarify in the text what "t" represents. Tendon thickness?

Equations 4 and 5, please clarify in the text what these equations are calculating. Also specify that x is in cm.

Values in methods section switch between cm and mm units. Please use one or the other consistently.

Equation 7, the terms or subscripts (i.e. Mm and Mj) should be defined for clarity. Similarly, define Ft for Equation 8.

Line 150, please provide a reference for "assumption that muscle volume is proportional to tendon force."

Tables 1 and 2 should be switched.

Validity of the findings

The study was limited by performing analysis in only one elbow posture, one that is arguably least functional but would seemingly result in the greatest change in moment arm with change in ACSA. It would be beneficial to show how moment arm changes with ACSA in a flexed elbow posture. Alternatively, please address this as a limitation in the text.

The main observations from the results presented in Figure 1 and Table 1 should be synthesized and presented in the Results section. For instance, how much of the increase in muscle-generated joint moment was attributed to an increase in ACSA alone versus an increase in moment arm? These findings are only presented in part C of Figure 1 but never mentioned in the Results or Discussion section. Your argument (e.g. line 202) that hytrophy/atrophy-related moment arm changes are important to consider when estimating muscle moment may be strengthened by distinguishing ACSA and moment arm effects on muscle moment.

Line 271, the reduction in normalized force with increasing ACSA was explained as "likely reflective of modeled increase in fascicle angle". Couldn't this be confirmed by manipulating variables within the model? Also, it seems that the decrease in normalized force, contrary to experimental findings, is drastic. In the end, the authors attribute the contradiction to not accounting for "central" or "peripheral" factors (line 279) in the model. However, the contradiction may be due to the several simplifying assumptions in building the model. More explanation is needed.

Line 317 states "reported numbers are not different from those in the literature", but the differences noted in the discussion section regarding moment contributions (lines 181-183) and normalized force (lines 277-278) seem to contradict the concluding statement.

Table 1 (as defined in manuscript) should have a row at least listing the collective joint moment contributions of other muscles to clarify the summation of all joint moments to 30 N-m.

Line 282-283, please specify whether increase in single fiber specific tension was associated with hypertrophy or atrophy.

---

## Round 0.2 · Major Revisions

The reviewers agree that the paper has improved appreciably but moderate revisions and clarifications are needed, including some scientific issues of substance that still need sorting out. However, if you provide a detailed point-by-point Response document and Tracked Changes MS when you submit your new version, I can check that and if I am convinced, I may accept the paper without further review.

·

Basic reporting

General comments:

The authors have adequately addressed my previous concerns and, with the exception of one point (listed in validity of the findings section), I only have small suggestions mainly related to grammatical errors and improving clarity.

Line 63: remove “cos” as it is not necessary
Line 65: the phrase including “… moments as a result the joint acting as a pivot.” Is unclear
Line 103: the phrase “of the proximal elbow flexors” is repeated.
Line 118: “the” before “Equation 1” should be removed
Line 131: “The” should be removed.
Line 215-216: The authors state that this is the first model investigating moment arm changes, but then seem to reference a previous study in lines 231-233 (Sugisaki et al, 2014) that did something similar. I found these sets of statements contradictory. Could the authors please add some detail to clarify how the current study is “first”?
Line 230: During the revision process, how this proportionality was obtained appears to have been lost. How was this proportionality arrived at? For example, a reader may assume that it was an equation fit to the study results. It would be better to specifically clarify how the authors obtained this equation and a few sentences explaining this in the discussion should be provided as this proportionality is a key finding of the current study.
Eq 10: is “l” defined previously? Or θ? Please double check that all equation symbols are defined in the text.
Lines 300-328: These limitations could actually be framed as a modelling strength. The authors have chosen not to include these phenomena. By doing, so the model avoids having to “tease out” what could be potentially confounding factors. The current manuscript is well written as is, but the authors may wish to consider adding a sentence or two showing how this choices could be a strength and not just a limitation.
Lines 330-334: Please see comment in validity section below. The authors have used a single joint configuration to propose a general proportionality relationship between muscle mass and moment arm length. Although I understand the authors rationale for the generalization and the relationship may be valid over many joint configurations, this study does not provide any concrete evidence for this generalization as it stands. I strongly suggest that this limitation is worded so that the reader clearly understands that, specifically, the proposed proportionality relationship is based on a single elbow configuration.
Fig 1: Figures and tables should be stand-alone. As such, I feel some additional detail should be provided in the figure caption. For example, what is x? If a reader were to glance at figures/captions without reading the manuscript text, it would be hard to know what some of the figures may represent. I suggest using a couple sentences to describe each figure in the caption.
Fig 2a: The use of the word “normal” is confusing. Maybe use “empirical” instead? Or some other word like “nominal”?
Fig 2c: I do not see any BIC data in this figure. Are these data identical to the BRA data? If so a note in the figure caption should be provided.

Experimental design

The authors have adequately addressed all of my previous concerns within the area of experimental design and the manuscript is now clear as to how the model was constructed and evaluated.

Validity of the findings

In my opinion, the revised manuscript is much improved and the validation steps are now clearly presented in such a manner that a reader can assess the model’s validity.
However, I still have one area of concern with regards to extrapolating the proportionality relationship found in this study to all joint configurations. While the authors believe that this proportionality can be extrapolated, I feel including a couple additional joint configurations to show this is the case would be very valuable and could greatly strengthen the main finding of the manuscript. However, I also feel that the manuscript, in its current form, now allows a reader to make a clear assessment of author’s statements. As a minimum alternative, I feel the authors should make sure that this limitation (one joint configuration) is placed such that it is the most prominent limitation in the discussion and should also be included in the concluding remarks.

Reviewer 2 ·

Basic reporting

Thank you to the authors for their response to my comments. The changes made by the reviewers greatly improved the manuscript. The flow of ideas in the Introduction section is somewhat confusing, partly because the motivation for certain discussion points (e.g. relationship between hypertrophy/atrophy, pennation angle, and tendon force as it relates to PCSA) was not clear. The following are specific comments about the Introduction section.

In the revised introduction, there is a discussion (paragraph 2) about ACSA and PCSA, and how pennation angle affects the muscle force transmitted to the tendon with respect to changes in PCSA. How this discussion relates to the study is not immediately clear to the reader. Since you evaluate the effect of hypertrophy/atrophy on pennation angle in the study, it may be helpful to indicate in the Introduction why you are evaluating it (i.e. the quantitative relationship between muscle size and pennation angle is unknown, etc.), especially since you state in the next paragraph that changes in pennation angle with "mechanical loading" have already been well documented.

In the last sentence in paragraph 2 of the Introduction, no specific reference is provided except in a group of references in the first sentence of the next paragraph. Also, the wording of the sentence is confusing and addresses changes in pennation angle both DURING loading and FOLLOWING loading.

At the beginning of paragraph 4 of the Introduction, MVIC force is distinguished from internal isometric muscle force. It seems that MVIC "force" pertains to joint moment, whereas isometric muscle "force" is a linear force. If your point is to make this distinction, then specify in the first sentence that the forces are not identical because they are measuring two different things, rather than not being identical because of experimental error, technique, etc. It's not clear since you call both things "force".

The stated purpose of the paper (last paragraph in Introduction) should be modified since changes in ACSA were used to evaluate changes in MA length, pennation angle, and joint moment contributions.

Experimental design

There is a typo in the sentence just after Equation 1, "of the proximal elbow flexors" is stated twice.

After Figure 1 insertion point, there is a sentence fragment "Under the assumption that the".

In equations 4 and 5, why does x range up to 11.2 cm?

Validity of the findings

It looks like muscle force at ACSA=22 cm^2 is about 680 or 690 N (Figure 2C). At the same ACSA, pennation angle for BIC is 30 degrees. Computing tendon force as in Equation 8, BIC force at ACSA=22cm^2 should be around 600 N on Figure 2D, but it looks to be 630 or 640 N. It also appears that, since BIC and BRA have the same muscle force but BIC has higher pennation angle, then BRA tendon force should be higher in Figure 2D.

In the last sentence of paragraph 3 in the Limitations section, one "loading" should be changed to "unloading".

In the conclusions you state that the model "has not been validated", though in the last sentence of the Methods section you state that it was validated by comparing outcomes to values reported in the literature. Please clarify.

---

## Round 0.3 · accepted · Accept

Very nicely handled; I am glad that the final reviews helped. I can accept now-- but on line 137 (Tracked Changes MS) consider re-wording the "Under the assumption that the" sentence as right now it does not quite make sense as written. Congratulations!